# PCOS Physiopathology and Vitamin D Deficiency: Biological Insights and Perspectives for Treatment

**DOI:** 10.3390/jcm11154509

**Published:** 2022-08-02

**Authors:** Giuseppe Morgante, Ilenia Darino, Amelia Spanò, Stefano Luisi, Alice Luddi, Paola Piomboni, Laura Governini, Vincenzo De Leo

**Affiliations:** Department of Molecular and Developmental Medicine, University of Siena, 53100 Siena, Italy; giuseppe.morgante@unisi.it (G.M.); ileniadarino@hotmail.it (I.D.); ameliaspano94@gmail.com (A.S.); stefano.luisi@unisi.it (S.L.); alice.luddi@unisi.it (A.L.); paola.piomboni@unisi.it (P.P.); vincenzo.deleo@unisi.it (V.D.L.)

**Keywords:** polycystic ovary syndrome (PCOS), vitamin D, insulin resistance, oxidative stress, fertility, supplementation

## Abstract

Recent literature has stressed the importance of vitamin D (VD) in polycystic ovary syndrome (PCOS). Women with PCOS are deficient in VD, particularly those with a higher weight. Hypovitaminosis is a risk factor for glucose intolerance, and reduced levels of VD is associated with insulin resistance and increased diabetes risk. Since women with PCOS and hirsutism seem to have lower levels of VD than women with PCOS without hirsutism, a correlation between VD deficiency and hyperandrogenism may be suggested. Interestingly, VD is crucial for many human physiological functions, including to counteract inflammation and oxidative stress. Some studies evaluated effects of VD supplementation on glucose homeostasis variables, hormonal status, lipid concentrations, and biomarkers of inflammation and oxidative stress among VD-deficient women. Moreover, VD has been shown to play a role in egg quality and fertility. This review aims to show the relationship between VD and the endocrine and metabolic profile of PCOS patients, as well as its implications for their fertility. The supplement of VD to the common therapy can lead to an improvement of the insulin resistance and lipid metabolism, a reduction of circulating androgens, as well as a better response to the induction of ovulation in PCOS women.

## 1. Introduction

Polycystic ovary syndrome (PCOS) constitutes the most frequent endocrine disorder in women of reproductive age. PCOS affects up to one-sixth of women with a prevalence that may reach or even exceed 10–15%, depending on the diagnostic criteria applied and the populations studied in different geographical areas [1,2]. It is a multisystem disorder characterized by oligo or anovulation, and consequently oligo or amenorrhea, and the development of hyperandrogenism, resulting from circulating luteinizing hormone (LH) levels and the altered ratio of LH to follicle stimulating hormone (FSH) [3]. Morphologically, the ovaries may appear polycystic (polycystic ovarian morphology, PCOM). PCOS is also associated with hyperinsulinemia, impaired glucose tolerance, and sometimes even type 2 diabetes mellitus (T2DM) [4]. Factors such as insulin resistance (IR) dyslipidemia, endothelial connection, and systemic inflammation are other elements that add to this set of signs and symptoms and predispose patients to a higher risk of cardiovascular disease than women without PCOS [5,6]. Probably, the reduced insulin sensitivity leads to an inevitable compensatory hyperinsulinemia, and this contributes to the development of hyperandrogenism through a chronic stimulus directed toward the cells of the ovarian theca. Evidence has suggested a correlation between IR pathogenesis and vitamin D (VD) deficiency, placing hypovitaminosis as a causal factor for the metabolic syndrome in PCOS women [7]. The vitamin D receptor (VDR) is almost ubiquitously expressed, regulating at about 3% of human genome, counting the genes coding for glucose metabolism as well [8]. This suggests the role and the correlation of VD deficiency with PCOS symptoms such as T2DM, IR, and cardiovascular diseases. A meta-analysis provided evidence regarding the correlation of VD receptor polymorphisms and PCOS, identifying some susceptibility markers as well. To date, the use of metformin in PCOS patients represents a milestone in therapy for these patients [9,10]. Supplementation with natural molecules, such as VD, may help overcome PCOS-related symptoms [11]. This review, analyzing data from the published literature on PubMed, identifies the different possible contribution of VD deficiency in the physiopathology of PCOS, and aims to show how VD supplementation to common therapy results in improved IR, reduced circulating androgens, and improved response to ovulation induction in women with PCOS.

### The Pathophysiology of PCOS

The 2003 Rotterdam workshop consensus, in accordance with the recommendations of the European Society of Human Reproduction and Embryology (ESHRE) [12], established criteria for diagnosis based on the association of at least two of three clinical features: (i) clinical or biochemical signs of androgen excess, such as acne, alopecia, and hirsutism; (ii) ovarian dysfunction with oligo-/anovulation; and (iii) polycystic ovaries on ultrasound examination.

These criteria were utilized to classify patients into four different phenotypes [13]: (i) phenotype A, the most prevalent, which is characterized by clinical and/or biochemical hyperandrogenism, menstrual dysfunction (oligo/amenorrhea), and ultrasonographic evidence of polycystic ovaries; (ii) phenotype B, characterized by hyperandrogenism, and menstrual dysfunction (oligo/amenorrhea); (iii) phenotype C is characterized by hyperandrogenism, and polycystic ovarian morphology; (iv) phenotype D, defined by oligomenorrhea, polycystic ovarian morphology, and normal androgens.

The exact etiology of PCOS remains poorly understood to date. The pathophysiological picture of PCOS is complex and involves several elements: the ovary, the hypothalamus, and genetic susceptibility and metabolic syndrome. Recently, IR and hyperandrogenism have assumed the role of key factors in the genesis of this disease [14]. Certainly, the first deficit is functional ovarian hyperandrogenism (FOH), caused by steroidogenic hyperactivity, which disrupts ovarian synthesis of both androgens and estrogens [15]. The main cause of FOH can be attributed to increased secretion by the hypothalamus of GnRH and subsequent release of LH by the pituitary gland. This improper secretion is manifested by an elevated LH/FSH ratio. Consequently, there will be increased thecal secretion of androgens, which is manifested by the presence of small and numerous growing antral follicles. These follicles are found to be more resistant to the hormonal activity of FSH and, consequently, the increased concentration of LH will inhibit the proliferation of granulosa cells, causing their premature luteinization. This abnormality associated with reduced sensitivity to FSH leads to blockage of follicular maturation, resulting in oligo-/anovulation [16]. Overstimulation of theca cells by LH is exacerbated by insulin, which acts directly through the insulin receptor or indirectly through the growth factor 1 (IGF-1) receptor [17].

## 2. Vitamin D

VD regulates calcium metabolism and bone mineralization. VD exists in two forms: ergocalciferol (VD2) and cholecalciferol (VD3). VD3 is of animal origin and is synthesized endogenously in the human body. Ultraviolet radiation from the sun’s rays acts on the skin and converts 7-dehydrocholesterol to a pre-vitamin form that is subsequently converted to VDFor VD activation, two different hydroxylations are required: the first occurs in the liver where the enzyme 25-hydroxylase metabolizes VD to 25-hydroxyvitamin D (25-OH-D). Subsequently, 25-OH-D is transported to the kidneys where it undergoes the second hydroxylation; thus 1,25-dihydroxyvitamin D (1,25-OH-D) or calcitriol, the biologically active form of VD, is formed [18].

It is estimated that there are 1 billion people who are VD-deficient or VD-insufficient in the world. The Endocrine Society of North America has defined VD deficiency as 25-OH-D levels < 20 ng/mL and insufficiency as 20–30 ng/mL. Numerous studies state with certainty that deficiency of this hormone is linked to IR, impaired glucose and lipid metabolism, and ultimately infertility—scenarios commonly seen in women with PCOS [19,20].

VDR is a transcription factor that, by eliciting the genomic actions of VD, is able to regulate several endocrine and cell functions including calcium metabolism. Both VD and calcium are known to be associated with endocrine dysfunctions, insulin resistance, and type 2 diabetes in PCOS [3].

All of this information, along with the reported key role of VDR in the regulation more than 3% of the human genome, prompted researchers to examine the real contribution of the VDR gene polymorphisms in metabolic and endocrine disturbance of PCOS [21,22]. Despite some evidence pointing to an influence of VDR gene variants in PCOS features, it is difficult to definitively establish a clear association of VDR polymorphisms with the development of PCOS [23].

### 2.1. Vitamin D Deficiency and PCOS Phenotypes

Interestingly, it has been shown that low VD levels may worsen PCOS symptoms, so that an inverse correlation has been reported between serum VD level and metabolic and hormonal disturbances of PCOS [7,20,24].

Few studies investigated the relationship between VD deficiency and PCOS phenotypes. Davis et al. analyzed PCOS women, grouped into three diagnostic phenotypes according to the Rotterdam criteria: (i) group 1, women with ovulatory dysfunction and polycystic ovaries; (ii) group 2, women with ovulatory dysfunction and androgen excess; (iii) group 3, women with ovulatory dysfunction associated with polycystic ovaries and androgen excess. According to this study, a higher prevalence of VD deficiency in PCOS cases with androgen excess may be demonstrated [25].

Maktabi et al. performed a placebo-controlled trial on VD-deficient (serum concentrations < 20 ng/mL) women with phenotype B-PCOS according to the Rotterdam criteria. After the 12-week intervention, VD supplementation significantly decreased fasting plasma glucose, insulin, HOMA-IR index, and increased quantitative insulin sensitivity [26].

The main mechanisms described in this review that have a key role in the direct and indirect activity of VD on female fertility are plotted in Figure 1.

### 2.2. Vitamin D, Hypothalamic–Pituitary–Gonadal Axis, and Androgen Levels

Despite numerous studies investigating the effects of VD on reproductive function and gonadal hormone production, our knowledge about the mechanism by which VD affects reproductive physiology is still limited.

Regarding to the impact of VD on the hypothalamic–pituitary–gonadal (HPG) axis, an interesting insight comes from mini-puberty, the period of time within the first few months of life during which a transient activation of the HPG axis correlates to a brief activation of gonadal hormone production [27].

This HPG activation is important for future gonadal function. According to Kılınç et al., there is an association between 25-OH-D levels and gonadal hormones at mini-puberty. Total testosterone level was higher and inhibin B was lower in 25OH-D deficient than sufficient girls, while a modest effect of 25OH-D was identified on total testosterone and inhibin B. Therefore, the 25OH-D seems to have an effect on gonadal function during early life [28].

In regard to androgen levels, a study demonstrated a positive relationship between serum VD level and total testosterone and free androgen index. Therefore, this finding lets us hypothesize that VD may improve female fertility by modulating androgenic activity [29]. This is not surprising, since VD is able to affect the expression and activities of some of the enzymes involved in the production of sex hormones [24,30,31] (see *Vitamin D, Ovarian Physiology and Oxidative Stress* Section).

Confirming this, numerous studies associated VD deficiency with an alteration in serum levels of dehydroepiandrosterone, testosterone, sex hormone binding globulin (SHBG), and free androgen. In particular, testosterone is reported to be significantly higher in patients with PCOS compared to non-PCOS controls. The effect of VD supplementation on regulating testosterone unbalance is reported by a pilot study [32], conducted on overweight women with PCOS and VD deficiency that have been supplemented with high doses of this vitamin and calcium daily. This treatment induced, after 3 months, a significant reduction in the total level of testosterone and androstenedione. This is a proof of concept of the direct effects of VD and calcium supplementation on the steroidogenesis pathway (ovarian and/or adrenal). Therefore, all these data suggest potential therapeutic benefits of VD and calcium supplementation in ameliorating hormonal milieu and PCOS related sequelae in women deficient in VD.

### 2.3. Vitamin D, Ovarian Physiology, and Oxidative Stress

Many studies have demonstrated that VD can alter anti-Mullerian hormone (AMH) signaling, follicle stimulating hormone (FSH) sensitivity, and progesterone (P) production and release in human granulosa cells, indicating a possible physiologic role for VD in ovarian follicular development and luteinization [33,34]. Indeed, in human luteinized granulosa cells, VD decreases the expression of both the AMH receptor and FSH receptor. Following follicular selection in a women’s late follicular phase, the follicle becomes less dependent on FSH and more dependent on LH, followed by terminal maturation and ovulation. AMH and FSH receptor expression in granulosa cells has been found to be the highest in small immature follicles and to diminishes gradually with the progression of the maturation of oocytes [35,36].

On the other hand, animal models have shown that VD stimulates ovarian steroidogenesis by inducing the expression of both dehydroepiandrosterone sulfotransferase (DHEAS), an enzyme that mediates sulfo-conjugation of endogenous hydroxysteroids, and of aromatase [37,38,39]. Moreover, VD has been also reported to increase in vitro 3β-hydroxysteroid dehydrogenase (3β-HSD) RNA levels, possibly reflecting a state of granulosa cells luteinization [39,40]. Finally, it has been reported that VD may change the expression of the aromatase, the enzyme catalyzing the biosynthesis of estrogen, which is an androgen precursor [29,41,42].

PCOS women show elevated levels of advanced glycation end products (AGEs), a pro-inflammatory molecule family. AGEs and their receptors may contribute to the pathogenesis of PCOS, with negative consequences on metabolic and reproductive fields. Data from the literature indicate that VD might improve the PCOS phenotype and could alleviate the detrimental effects of AGEs [43]. According to these data, VD may play a pivotal role in enhancing key steroidogenic enzymes, thus potentiating granulosa cell luteinization and providing a better ovarian environment [44].

VD is an essential antioxidant because of its ability to control systemic inflammation, oxidative stress, and mitochondrial respiratory function in humans [45]. Interestingly, various studies reported the presence of oxidative stress in PCOS patients. Indeed, key oxidative stress markers such as malondialdehyde (MDA), nitric oxide (NO) advanced glycosylated end products (AGEs), and xanthine oxidase are increased in PCOS patients [46]. Moreover, mitochondrial dysfunction, along with the correlated increased ROS production, explains the oxidative status in PCOS patients, even if it should be mentioned that the oxidative status varied between individuals because of changes in lifestyle, diet, and antioxidant uptake.

In this context, the well described antioxidant activity of VD seems to play a pivotal role. In the presence of physiologic concentration of VD, nuclear factor-E2-related factor 2 (Nrf2) transcription is activated through VDRE, and then Nrf2 translocates from the cytoplasm to the nucleus, thus activating the expression of several genes with antioxidant activity [45]. Therefore, low levels of VD are correlated to decreased Nrf2 transcription and, in turn, to increased risks from oxidative stress-related tissue damage.

### 2.4. Vitamin D, Insulin Resistance, and Obesity

Insulin resistance (IR) is one of the more specific traits of PCOS, and is mainly marked in obese women, suggesting that PCOS and obesity have a synergistic effect on the magnitude of the insulin disorder, leading to increased insulin secretion by pancreatic β-cells and compensatory hyperinsulinemia [47]. IR and related hyperinsulinemia have been linked to all symptoms of the syndrome, such as reproductive disorders, hyperandrogenism, acne, hirsutism, and metabolic disturbances. Finally, insulin resistance in PCOS may be considered a risk factor for gestational diabetes [48]. To this regard, one theory relies on the regulatory effect of VD on the intracellular and extracellular calcium level that is essential for insulin-mediated intracellular processes and may have impact on insulin secretion [49,50,51,52]. Another hypothesis involves the stimulatory effect of VD on the expression of insulin receptors leading to the increase of insulin sensitivity. Indeed, VD activates the transcription of the VD response element (VDRE) of the human insulin gene [53].

In the clinical practice, many studies stressed that serum 25OH-D is negatively correlated with body mass index (BMI) [54,55,56,57]. In women with type 2 diabetes mellitus, an association between low levels of VD and increased insulin resistance was found [58].

There is some proof suggesting that VD deficiency might be involved in the pathogenesis of insulin resistance in PCOS [59,60]. VD deficiency is a contributing factor to IR, obesity, and metabolic syndrome, all of which are commonly associated with ovulatory dysfunction; indeed, a VD supplement implies a better and healthier ovarian physiology [61,62,63].

In any case, what may be the mechanisms underlying the association of low 25OH-D levels and insulin resistance is still a matter of debate. It is of note that the association of obesity with VD deficiency warrants more discussion, since it is not well understood if VD deficiency comes from obesity (e.g., VD may be trapped in fat tissues) and/or if obesity is due to VD insufficiency [64].

Ott et al. investigated the correlation between serum 25OH-D concentrations and metabolic parameters in obese and non-obese women with PCOS [65]. The serum 25OH-D mean levels were lower in obese PCOS patients. There was an association of increased HOMA-IR, BMI, triglycerides, and total testosterone, with decreased 25OH-D concentrations in obese PCOS patients.

Gallea et al. evaluated the role of body weight on the serum VD levels in women with or without PCOS. Results show that VD levels were lower in obese PCOS, and VD serum levels were comparable between normoinsulinemic lean PCOS women and controls. In conclusion, weight and hyperinsulinemia had a significant influence on these values [66].

More recently, one study has showed that VD supplement in obese female and deficient adolescents was associated with significant rising in insulin sensitivity [63]. After 6 months, although there were no significant differences between groups in BMI, serum inflammatory markers, and plasma glucose concentrations, women supplemented with VD had increased serum 25OH-D concentrations and significantly reduced HOMA-IR (marker of insulin resistance) and fasting plasma insulin. According to these results, the association of low VD levels with insulin resistance might, at least in part, be mediated by obesity.

However, the association of VD deficiency with insulin resistance may also be explained by mechanisms other than obesity. First of all, it has been reported that VD may improve insulin activity by stimulating VDR gene expression. Indeed, it is known that VDR is present in the promoter of the human insulin gene [67], and that 1,25OH-D3 is able to induce the transcription of the human insulin gene [53]. Moreover, VD may affect insulin-responsiveness through calcium, since insulin secretion is a calcium-dependent process [51]. Finally, by means of its immunomodulatory effect [68], low levels of VD may induce an inflammatory response, which is again associated with insulin resistance [69].

## 3. Relevance of Supplementation

### 3.1. Impact of Vitamin D Supplementation on Biomarkers of Oxidative Stress in PCOS

The effectiveness of VD supplementation on the reduction of biomarkers of inflammation and oxidative stress among women with polycystic ovary syndrome is controversial (Figure 2).

A recent systematic review and meta-analysis of randomized controlled trials provided evidence for a significant improvement in high-sensitivity C-reactive protein, MDA, and total antioxidant capacity in women with PCOS receiving a VD supplementation, while no effects were reported for both nitric oxide (NO) and glutathione (GSH) levels [70].

Others studies have reported that VD showed benefit in improving oxidative stress [26,71]. A randomized double-blind placebo-controlled clinical trial involving 104 overweight VD-deficient PCOS women were randomly supplemented with 1000 mg calcium daily and/or 50,000 IU VD weekly for 8 weeks. The combined calcium plus VD supplements had greater decreases in plasma MDA concentrations, and significant increases in plasma total antioxidant capacity and GSH levels compared with calcium alone, VD alone, and placebo groups, suggesting that calcium plus VD co-supplementation had beneficial effects on inflammatory factors and biomarkers of oxidative stress [71]. Interestingly, the co-administration of VD and probiotic for 12 weeks to women with PCOS was demonstrated to significantly improve the serum levels of high-sensitivity C-reactive protein, plasma total antioxidant capacity, GSH, and MDA [72].

Despite these findings, others studies did not observe a beneficial effect from VD supplementation [73]. Further studies are needed to define the effects of only VD on biomarkers of inflammation and oxidative stress.

### 3.2. Vitamin D Supplementation and Fertility Outcomes in PCOS Women

In PCOS women, VD is related to menstrual irregularity, altered follicular development, ovulatory dysfunction, metabolic alterations, and decreased pregnancy rate [34,74,75,76].

In a prospective cohort study, the parathyroid hormone (PTH), the active form 1,25-hydroxy vitamin D3 (1,25OH-D3), and testosterone were measured in infertile women with PCOS undergoing clomiphene citrate stimulation [65]. This study demonstrated that high PTH hormone levels correlated with low serum calcium and low 1,25OH-D3, whereas high PTH was associated with high body mass index and with higher testosterone serum levels. When comparing women who had developed a follicle with those who were resistant after stimulation with 50 mg of clomiphene citrate, lower 1,25OH-D3 serum levels were detected in resistant ones. Moreover, significantly improved pregnancy rate was highlighted in women with higher BMI and lower 1,25OH-D3 serum levels. Finally, the significant correlation between lower 1,25OH-D3 serum levels and lower follicle development after stimulation with 50 mg of clomiphene citrate may be ascribed to the well reported role of 1,25OH-D3 in ovarian activity.

The effects of calcium-VD and metformin supplementation on the menstrual cycle and ovulation of patients with PCOS was investigated in a randomized clinical trial, enrolling 60 infertile PCOS patients [74]. Menstrual regularity and the number of dominant follicles (≥14 mm) during the 2–3 months of follow-up was higher in the calcium-vitamin D plus metformin group than in either of the other two groups.

Fang et al. evaluated the effect of VD supplementation on patients with PCOS. According to this review, VD supplementation significantly improves follicular development with a higher number of dominant follicles. Moreover, the combined supplementation with metformin plus VD improves the regularity of the menstrual cycles [77].

There is evidence suggesting that 25OH-D may also play an important role during pregnancy, but data regarding VD deficiency during gestation in PCOS patients and its association with perinatal outcome is limited. It is already well known that vitamin supplementation is safe and improves VD and calcium status; also, during pregnancy, sufficient 25(OH)D supplementation can prevent neonatal hypocalcemia, which may result in the softening of bones. VD supplementation decreased the risk of babies being small for gestational age and increased birth weight [78]. Therefore, these observations enable us to hypothesize the beneficial effects of VD supplementation in patients with PCOS seeking a pregnancy.

It is of note that a relationship exists between VD and markers of ovarian reserve. AMH is a glycoprotein produced by granulosa cells of primary follicles in the ovaries and then secreted in the blood and is considered to be one of the best ovarian reserve markers [33,79]; its expression and serum levels are altered by environmental factors, such as VD deficiency and obesity. The non-significant fluctuation of AMH during the menstrual cycle represents a strength over the other ovarian reserve markers (such as day 3 FSH), making it clinically useful and convenient. Investigators identified a functional VDRE in the promoter region of the human AMH gene, which demonstrates a potential direct effect of VD on AMH expression [80].

In the serum, 25OH-D is positively correlated with AMH, and appropriate VD supplementation in VD-depleted women can suppress the seasonal changes that occur in serum AMH. In VD-deficient women with PCOS, VD supplementation lowers the abnormally elevated serum AMH levels, possibly indicating a mechanism by which VD improves folliculogenesis [33,81].

The huge number of observational studies described in the following paragraphs of this review, shedding light on the association between VD status with PCOS, are summarized in Table 1.

## 4. Conclusions

VD has protective effects on the cardiovascular system and on chronic and autoimmune diseases, regulates the expression of genes involved in glucose and lipid metabolism, and plays a key role in the reproductive system of women [6,45,83]. VD deficiency has been shown to be associated with many of the signs present in PCOS: ovulatory dysfunction, hyperandrogenism, insulin resistance, diabetes and dyslipidemia, adiposity indices, and systemic proinflammatory environments [25,64,84].

In general, PCOS women have lower 25-OH-D levels compared to healthy controls, even if the mechanisms involved in this dysfunction are still debated. In addition, association exists between VD levels and obesity. Moreover, despite conflicting results, this review highlighted an increased oxidative stress level in women with PCOS.

The in-depth knowledge of these mechanisms might potentially lead to this oral, rather safe and cost-effective vitamin becoming an adjunct treatment in therapies for PCOS patients. To this regard, in the clinical management of PCOS patients, the measurement of VD serum levels, along with other endocrine markers and the patient’s phospho-calcium metabolism, should be always recommended. This will lead the clinician to evaluate the oral VD dose necessary, in association with other more specific therapies. This approach may be an effective weapon in obese women and in the treatment of PCOS patients with insulin-resistance. Excluding kidney, liver, or internistic disease that modifies absorption, after evaluation of VD3 dose requirement, data from the literature suggest that the dose of 1000 IU per day corresponding to 25 mcg seems to be the most effective at raising 25-OH-D levels to sufficient amounts, during three months of therapy in PCOS women with VD deficiency [85].

Future contributions of VD should have broad significance to increase natural conception in women with PCOS with an aim to decrease costs related to in vitro fertilization procedures. In conclusion, based on these data, we suggest that VD administration in PCOS women can represent a safe strategy to improve their symptoms without adverse effects.

## Figures and Tables

**Figure 1 jcm-11-04509-f001:**
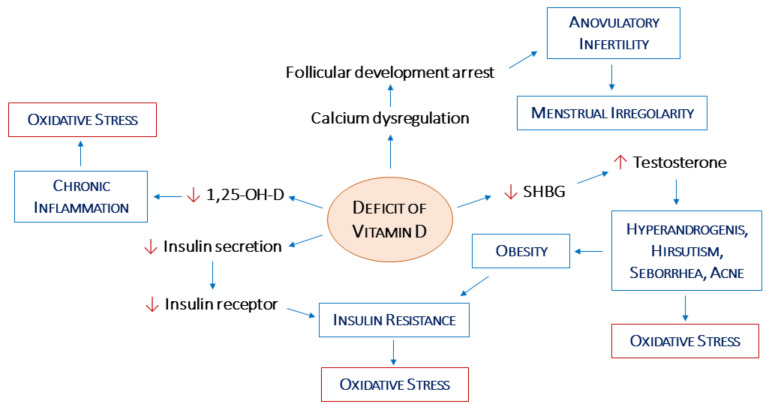
Relationship of vitamin D deficiency with the pathogenesis of insulin resistance and the metabolic syndrome in PCOS, hormonal alteration, and infertility. (1,25-OH-D: 1,25-hydroxyvitamin D; SHBG: Sex Hormone Binding Globulin).

**Figure 2 jcm-11-04509-f002:**
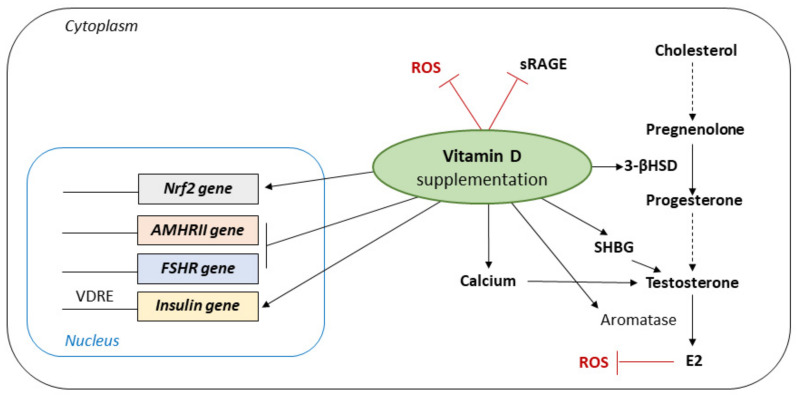
Impact of vitamin D supplementation on ovarian cells physiology. (Nrf2: Nuclear factor erythroid 2-related factor 2; AMHRII: Anti-Müllerian Hormone Receptor type 2; FSHR: Follicle Stimulating Hormone Receptor; VDRE: Vitamin D Response Element; ROS: Reactive Oxygen Species; sRAGE: Soluble Form of the Receptor for Advanced Glycation Endproduct; 3-βHSD: 3β-Hydroxysteroid dehydrogenase; SHBG: Sex Hormone Binding Globulin; E2: Estradiol).

**Table 1 jcm-11-04509-t001:** Summary of observational studies.

*Issue*	*References*	*Main Outcomes*
** *Vitamin D deficienc and PCOS phenotypes* **	[7,20,24]	VD low levels may worsen PCOS symptoms; an inverse correlation has been reported between serum VD level and metabolic and hormonal disturbances in different PCOS phenotypes.
	[25,26]	A higher prevalence of VD deficiency in PCOS cases with androgen excess may be demonstrated. In the B-PCOS phenotype, VD supplementation significantly decreased fasting plasma glucose, insulin, HOMA-IR index, and increased quantitative insulin sensitivity.
** *Vitamin D, PG axis, and androgen levels* **	[29]	VD may improve female fertility by modulating androgenic activity.
	[28]	Association between 25-OH-D levels and gonadal hormones at mini-puberty.
	[30,31]	VD is able to affect the expression and activities of some of the enzymes involved in the production of sex hormones.
	[32]	Effect of VD supplementation on regulating testosterone unbalance.
** *Vitamin D, ovarian physiology, and oxidative stress* **	[46]	Oxidative stress markers are increased in PCOS patients.
	[45]	Low levels of VD are correlated to decreased Nrf2 transcription and increased risks from oxidative stress-related tissue damage.
	[33,34]	VD shows a possible physiologic role in ovarian follicular development and luteinization, and VD supplement can contribute to these processes.
	[35,36]	In human luteinized granulosa cells, VD decreases the expression of both the AMH receptor and FSH receptor.
	[37,39]	VD stimulates ovarian steroidogenesis by inducing the expression of DHEAS and aromatase.
	[40,39]	VD increases the 3β-HSD RNA levels in vitro, possibly reflecting a state of granulosa cell luteinization.
	[29,41,42]	VD may change the expression of the aromatase, the enzyme catalyzing the biosynthesis of estrogen, which is an androgen precursor.
** *Vitamin D, insulin resistance, and obesity* **	[61,62,63]	VD deficiency is a contributing factor to IR, obesity, and metabolic syndrome, all of which are commonly associated with ovulatory dysfunction: a VD supplement implies a better and healthier ovarian physiology.
	[64]	Association between concentration of VD and obesity has been strongly demonstrated both in adults and in adolescents: adipose tissue decreases circulating 25OH-D by trapping it.
	[65]	Association of increased HOMA-IR, BMI, triglycerides, and total testosterone, with decreased 25OH-D concentrations in the obese PCOS patients.
	[66]	Weight and hyperinsulinemia had a significant influence on these values: VD levels were lower in obese PCOS women, and VD serum levels were comparable between normoinsulinemic PCOS women and controls.
	[53,67]	VD may improve insulin activity by stimulating VDR gene expression. VDR is present in the promoter of the human insulin gene and 1,25OH-D3 is able to induce the transcription of the human insulin gene.
	[51]	VD may affect insulin-responsiveness through calcium, since insulin secretion is a calcium-dependent process.
	[68,69]	VD shows an immunomodulatory effect: low levels of VD may induce an inflammatory response, which is associated with insulin resistance.
** *Vitamin D supplementation and oxidative stress in PCOS* **	[70]	Significant improvement in high-sensitivity C-reactive protein, MDA, and total antioxidant capacity in women with PCOS receiving VD supplementation, while no effects were reported for NO and GSH levels.
	[71]	The combined calcium plus VD supplements had greater decreases in plasma MDA concentrations, and significant increases in plasma total antioxidant capacity and GSH levels compared with calcium alone, VD alone, and placebo groups.
	[72]	Co-administration of VD and probiotic to women with PCOS significantly improved the serum levels of high-sensitivity C-reactive protein, plasma total antioxidant capacity, GSH, and MDA.
	[73]	Beneficial effects from VD supplementation were not observed.
** *Vitamin D and fertility outcomes in PCOS women* **	[74,76]	VD is related to menstrual irregularity, altered follicular development, ovulatory dysfunction, metabolic alterations, and decreased pregnancy rate.
	[82]	Significant improvement in regulating menstrual abnormalities and follicle maturation in women receiving calcium and VD supplementation.
	[77]	The combined supplementation with metformin plus VD improves the regularity of the menstrual cycles.
	[33,79]	Relationship between VD and AMH: its expression and serum levels are altered by environmental factors, such as VD deficiency and obesity. In the serum, 25OH-D is positively correlated with AMH, and appropriate VD supplementation in VD-depleted women can suppress the seasonal changes that occur in serum AMH. In VD-deficient women with PCOS, VD supplementation lowers the abnormally elevated serum AMH levels, possibly indicating a mechanism by which VD improves folliculogenesis.

## Data Availability

Not applicable.

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
