# Peer review of "PCOS Physiopathology and Vitamin D Deficiency: Biological Insights and Perspectives for Treatment"

_jcm, 2022, doi:10.3390/jcm11154509_

Round 1

Reviewer 1 Report

The manuscript entitled “PCOS physiopathology and Vitamin D deficiency: biological 2 insights and perspectives for treatment” by Giuseppe Morgante et al. undertakes a very interesting problem related to the leading endocrinopathy in premenopausal women. The question of the Vitamin D deficiency seems to be a relatively new and promising direction in the research of PCOS physiopathology and its therapy. I really appreciate the Authors’ summary provided in Table 1.

In my opinion the work can reach a high scientific interest, the following minor issues should be addressed:

The manuscript is well-written and reasonably structured, and each section and subsection describes a selected aspect of the topic in a detailed way. Nevertheless, in my opinion some citations are missing.

The extended discussion of the attenuating effects of vitamin D for AGEs in women with PCOS could be recommended. Please look at the following works of Merhi et al.

Merhi, Z.; Buyuk, E.; Cipolla, M. Advanced glycation end products alter steroidogenic gene expression by granulosa cells: An effect partially reversible by vitamin D. MHR Basic Sci. Reprod. Med. 2018, 24, 318–326

Merhi, Z. Crosstalk between advanced glycation end products and vitamin D: A compelling paradigm for the treatment of ovarian dysfunction in PCOS. Mol. Cell. Endocrinol. 2019, 479, 20–26.

Beside the ref 47, the newer work of Lerchbaum et al. could be also cited (which focuses on the female fertility):

Lerchbaum, E.; Rabe, T. Vitamin D and female fertility. Curr. Opin. Obstet. Gynecol. 2014, 26, 145–150

In this review the Authors consider for example the greater chances of getting pregnant for the PCOS women with a normal Vit D level.

The advantageous effects of vitamin D supplementation in PCOS women was recently also postulated in the review:

Wawrzkiewicz-Jałowiecka, A.; Kowalczyk, K.; Trybek, P.; Jarosz, T.; Radosz, P.; Setlak, M.; Madej, P. In Search of New Therapeutics—Molecular Aspects of the PCOS Pathophysiology: Genetics, Hormones, Metabolism and Beyond. Int. J. Mol. Sci. 2020, 21(19), 7054.

It is an optional extension of the work. Namely, it is an interesting phenomenon that the PCOS patients are at elevated risk of VD deficiency, but if the PCOS patients are successfully getting pregnant, there are no statistically significant differences between the perinatal VD levels in PCOS and non-PCOS women. (Kollmann, Martina, et al. Vitamin D Concentrations at Term Do Not Differ in Newborns and Their Mothers with and without Polycystic Ovary Syndrome. J. Clin. Med. 2021, 10.3: 537.) The Authors can optionally add an appropriate comment.

Additional remarks:

In a sentence: “It is estimated that there are 1 billion deficient or insufficient people in the world.” a word indicating Vitamin D is missing. It would be more precise, if it were written something like e.g. It is estimated that there are 1 billion people who are VD-deficient or VD-insufficient in the world.

The appropriate reference should be added in a sentence:

“[...] the dose of 1000 IU per day corresponding to 25 mcg seems to be the most effective at raising 25-OH-D levels to sufficient, during three months of therapy in women with VD deficiency”

Author Response

Thanks for your important comments. We modified our work in accordance with your suggestions.

Q1. The extended discussion of the attenuating effects of vitamin D for AGEs in women with PCOS could be recommended. Please look at the following works of Merhi et al.

- Merhi, Z.; Buyuk, E.; Cipolla, M. Advanced glycation end products alter steroidogenic gene expression by granulosa cells: An effect partially reversible by vitamin D. MHR Basic Sci. Reprod. Med. 2018, 24, 318–326

- Merhi, Z. Crosstalk between advanced glycation end products and vitamin D: A compelling paradigm for the treatment of ovarian dysfunction in PCOS. Mol. Cell. Endocrinol. 2019, 479, 20–26.

A1. Done. Added paragraph (line 200-204) and references.

Q2. Beside the ref 47, the newer work of Lerchbaum et al. could be also cited (which focuses on the female fertility): Lerchbaum, E.; Rabe, T. Vitamin D and female fertility. Curr. Opin. Obstet. Gynecol. 2014, 26, 145–150

A2. Done

Q3. In this review the Authors consider for example the greater chances of getting pregnant for the PCOS women with a normal Vit D level. The advantageous effects of vitamin D supplementation in PCOS women was recently also postulated in the review:

Wawrzkiewicz-Jałowiecka, A.; Kowalczyk, K.; Trybek, P.; Jarosz, T.; Radosz, P.; Setlak, M.; Madej, P. In Search of New Therapeutics—Molecular Aspects of the PCOS Pathophysiology: Genetics, Hormones, Metabolism and Beyond. Int. J. Mol. Sci. 2020, 21(19), 7054.

It is an optional extension of the work. Namely, it is an interesting phenomenon that the PCOS patients are at elevated risk of VD deficiency, but if the PCOS patients are successfully getting pregnant, there are no statistically significant differences between the perinatal VD levels in PCOS and non-PCOS women. (Kollmann, Martina, et al. Vitamin D Concentrations at Term Do Not Differ in Newborns and Their Mothers with and without Polycystic Ovary Syndrome. J. Clin. Med. 2021, 10.3: 537.) The Authors can optionally add an appropriate comment.

A3. Done. Added paragraph (line 342-349) and reference.

Additional remarks:

Q4. In a sentence: “It is estimated that there are 1 billion deficient or insufficient people in the world.” a word indicating Vitamin D is missing. It would be more precise, if it were written something like e.g. It is estimated that there are 1 billion people who are VD-deficient or VD-insufficient in the world.

A4. Done

Q5. The appropriate reference should be added in a sentence:

“[...] the dose of 1000 IU per day corresponding to 25 mcg seems to be the most effective at raising 25-OH-D levels to sufficient, during three months of therapy in women with VD deficiency”

A5. Done. Entered reference: Bacha et al; Vitamin D3 Dose Requirement That Raises 25-Hydroxyvitamin D to Desirable Level in Overweight and Obese Elderly. J Clin Endocrinol Metab. 2021 Aug 18;106(9):e3644-e3654. doi: 10.1210/clinem/dgab296.

Reviewer 2 Report

The article under the title: "PCOS physiopathology and Vitamin D deficiency: biological insights and perspectives for treatment", is a very interesting review that aims to show the relationship between Vitamin D level and endocrine and metabolic profile of PCOS female patients as well as its implications for their fertility.

I have, however, some minor concerns about the design that needs to be addressed.

My comments are listed below.

Abstract

p. 1, line13

The Authors need to clarify the phrase “carbohydrate intolerance” and describe more precisely or otherwise the sentence: "carbohydrate intolerance which is the first step towards insulin resistance"

Introduction

In the Introduction Section, the Authors need to clarify what databases they have used.

p.2, line 59

I wonder if the “Diagnosis of PCOS” should be a separate section. Please consider including this section in the Introduction.

Conclusions

The Conclusion section is confusing and does not fully correspond to the purpose of the article. The Authors need to clarify and shorten the text.

Author Response

Thanks for your important comments. We modified our work in accordance with your suggestions.

Q1. Abstract (p. 1, line13)

The Authors need to clarify the phrase “carbohydrate intolerance” and describe more precisely or otherwise the sentence: "carbohydrate intolerance which is the first step towards insulin resistance"

A1. The sentence has been edited to make it clearer.

Q2. Introduction

In the Introduction Section, the Authors need to clarify what databases they have used.

A2. Done

Q3. (p.2, line 59) I wonder if the “Diagnosis of PCOS” should be a separate section. Please consider including this section in the Introduction.

A3. Done

Q3. Conclusions

The Conclusion section is confusing and does not fully correspond to the purpose of the article. The Authors need to clarify and shorten the text.

A3. The authors revised the conclusions by reducing and clarifying them.

Reviewer 3 Report

The study is well written and offers a good overview about the topic.

A recently published study of the Journal of clinical medicine from the study group of Ott et al. who is cited about letrozole and vitamin D and calcium is missing. 

Author Response

Q1. A recently published study of the Journal of clinical medicine from the study group of Ott et al. who is cited about letrozole and vitamin D and calcium is missing.

A1. Done. Entered reference